# Genetic Analysis of *Candida albicans* Filamentation by the Iron Chelator BPS Reveals a Role for a Conserved Kinase—WD40 Protein Pair

**DOI:** 10.3390/jof10010083

**Published:** 2024-01-22

**Authors:** Mariel Pinsky, Daniel Kornitzer

**Affiliations:** Department of Molecular Microbiology, B. Rappaport Faculty of Medicine, Technion—I.I.T., Haifa 31096, Israel; mariel.ojtt@gmail.com

**Keywords:** morphogenesis, Yak1 kinase, forward genetics, *Candida albicans*, bathophenanthroline disulfonate, iron utilization

## Abstract

*Candida albicans* is a major human pathogenic fungus that is distinguished by its capability to switch from a yeast to a hyphal morphology under different conditions. Here, we analyze the cellular effects of high concentrations of the iron chelator bathophenanthroline disulfonate (BPS). BPS inhibits cellular growth by withholding iron, but when iron chelation is overcome by the addition of hemoglobin as an iron source, the cells resume growth as hyphae. The BPS hyphal induction pathway was characterized by identifying the hyphal-specific transcription factors that it requires and by a forward genetic screen for mutants that fail to form hyphae in BPS using a transposon library generated in a haploid strain. Among the mutants identified are the DYRK1-like kinase Yak1 and Orf19.384, a homolog of the DYRK1-associated protein WDR68/DCAF7. Orf19.384 nuclear localization depends on Yak1, similar to their mammalian counterparts. We identified the hyphal suppressor transcription factor Sfl1 as a candidate target of Yak1-Orf19.384 and show that Sfl1 modification is similarly affected in the *yak1* and *orf19.384* mutant strains. These results suggest that DYRK1/Yak1 and WDR68/Orf19.384 represent a conserved protein pair that regulates cell differentiation from fungi to animals.

## 1. Introduction

*Candida albicans* is a commensal organism of humans as well as a major opportunistic pathogen, capable of causing superficial infections among susceptible populations and deep-seated infections in immunosuppressed patients [1]. One of the best-studied traits of *C. albicans* is its ability to switch between multiple morphologies, including yeast, pseudohyphae, and hyphae [2]. This ability contributes to *C. albicans’* ability to cause disease in the susceptible host [3,4,5]. Many stimuli are known to induce the transition from yeast to hyphal growth, including external conditions such as an elevated temperature, the presence of serum, certain growth media formulations, N-acetylglucosamine, neutral to slightly alkaline pH, and interference with internal homeostatic regulations, such as cell cycle progression [6,7,8,9]. Here, we identified a new such stimulus, exposure to the iron chelator bathophenanthroline disulfonate (BPS) [10], and identified the conditions under which BPS induces filamentation. 

*C. albicans* is a diploid organism lacking a complete sexual cycle. As a consequence, using forward genetics to study the biology of this organism has been precluded by the fact that most random mutations usually yield recessive loss-of-function mutants. Genetic analysis in *C. albicans* has therefore traditionally relied on reverse genetics: genes identified by bioinformatics or, occasionally, by screening of *C. albicans* plasmid libraries in *S. cerevisiae* are deleted by homologous recombination, one allele sequentially after the other [11]. Large-scale targeted gene deletion libraries have been constructed over the years and screened for various phenotypes, e.g., [12,13], but none of these libraries cover the entire *C. albicans* genome. The recent identification in a haploid line of *C. albicans* has, however, opened the prospect of forward genetic analysis in this organism [14]. Transposon mutagenesis systems have been established that enable rapid construction, screening, and analysis of mutant collections [15,16,17].

Here, we used a combination of mutant library screening and transposon mutagenesis to isolate mutants that are not responsive to BPS-induced filamentation. Among the genes identified are those encoding a set of known filamentation-related transcription factors, as well as the kinase Yak1 and the WD40 repeat protein Orf19.384. 

## 2. Materials and Methods

### 2.1. Media and Chemicals

Cells were grown in YPD medium (1% yeast extract, 2% bacto-peptone, 2% glucose, and tryptophan at 150 mg/L) or in Synthetic Complete (SC) medium lacking specific amino acids, as indicated. SC medium contains, per liter, Yeast Nitrogen Base (USBiological, Salem, MA, USA) 1.7 g, (NH_4_)_2_SO_4_ 5 g, the 20 amino acids, adenine and uridine, 0.1 g each, except leucine, 0.2 g, glucose, 20 g, and 0.2 mM inositol. Media were supplemented with the ion chelators ferrozine or bathophenanthroline sulfonate (BPS) or with the siderophore ferrichrome at the indicated concentrations or bovine hemoglobin (all from Sigma Aldrich, St. Louis, MO, USA) from a 0.5 mM stock in Dulbecco’s phosphate-buffered saline (Biological Industries, Beit Haemek, Israel). 

### 2.2. Plasmids and Strains 

Plasmids KB2734 and KB2735 contain the *YAK1* 5′ region (−400 to +80, primers 9 + 10 in Appendix A, SacI-SpeI) and the 3′ region (2370 to 3047, primers 11 + 12, PstI-XhoI) at the two ends of the hisG-*URA3*-hisG “blaster” of plasmids KB985 and KB986, respectively [18,19]. KB2736 and KB2737 similarly contain the *ORF19.384* 5′ region (−273 to +80, primers 15 + 16, SacI-SpeI) and 3′ region (1607 to 2096, primers 17 + 18, PstI-XhoI) in plasmids KB985 and KB986, respectively. Plasmid KB2726 contains the *YAK1* open reading frame (610 to 2427) fused to the CaGFPgamma ORF [20] (primers 7 + 8), and KB2752 contains the *ORF19.384* ORF (302 to 1674) fused likewise (primers 13 + 14) [19]. 

The strains are listed in Table 1. KC1337 was generated by deleting the 1st allele of *YAK1* with plasmid KB2735, followed by treatment with 5-fluoroorotic acid (5FOA) to select for cells that have ejected the *URA3* marker. KC1339 was generated by transforming KC1337 with KB2734, followed by 5FOA treatment. KC1341 was generated by deleting the 1st allele of *ORF19.384* with plasmid KB2736, followed by treatment with 5FOA. KC1381 was generated by transforming KC1341 with KB2737, followed by 5FOA treatment. KC1543, KC1544, KC1545 were generated by introducing the 13xMyc tag after the *SFL1* open reading frame in KC274, KC1339, and KC1381, respectively, using a PCR product amplified from plasmid KB1541 [18] with primers 23, 24 (Appendix A). KC1669 and KC1672 are KC274 and KC1381 transformed with KB2726 digested with BSP119I; KC1670 and KC1671 are KC274 and KC1339 transformed with KB2752 digested with HpaI.

### 2.3. Growth Assays 

Overnight starter cultures grown in YPD were diluted into a series of two-fold dilutions of hemoglobin, or BPS, in the indicated media. Cells were inoculated in flat-bottomed 96-well plates at OD_600_ = 0.00001, 150 µL per well. Plates were incubated at 30 °C on an orbital shaker at 60 rpm, and growth was measured by optical density (OD_600_) after 1, 2, and 3 days with an ELISA reader. Cells were resuspended with a multi-pipettor before each reading. Each culture was performed in triplicate. 

### 2.4. Enrichment and Screening for Filamentation-Defective Mutants

A starting mutant pool a haploid *C. albicans* strain mutagenized by random transposon insertion [16] was enriched for non-filamenting mutants as described [19]. Briefly, cells were grown for 24 h in 5 mL YPD + 2 mM BPS and 0.25 µM hemoglobin at 30° while shaking, and then the cultures were left to sediment in a test tube on the assumption that non-filamenting mutants would sediment slower than the hyphal wild-type cells. After 10 min, the top 1 mL of the tube was removed and diluted in fresh 5 mL of medium and left to sediment again, after which the top 1 mL was diluted in fresh medium and grown for 24 h as before. This procedure was repeated 10 times. Finally, the enriched pool was plated on YPD plates and 2 mM BPS and 1 µM hemoglobin, incubated or 24 h, and the colonies (200/plate) were visually scanned with a binocular microscope for reduced hyphal formation.

### 2.5. Identification of Transposon Insertion Sites 

Single colonies were grown overnight in YPD medium. DNA was extracted as described in [27], and subjected to the FPNI DNA amplification protocol [28] using Ds-specific primers (Appendix A, primers 1–6), as described in [17]. PCR reactions were tested for the presence of bands on an agarose gel, and the positive reaction (>90%) was cleaned using the GenElute kit (Sigma-Aldrich) and sequenced by Sanger sequencing. The sequence was used to determine the position and orientation of the Ds insertion in the *C. albicans* genome. 

### 2.6. Microscopy

A Zeiss AxioImager M1 microscope (Carl Zeiss AG, Oberkochen, Germany) equipped with a Colibri 5 laser light source for epifluorescence was used throughout. For regular light microscopy, a 10× A-Plan objective or a 40× Plan-Neofluar objective with DIC optics were used. For visualization of fluorescently labeled cells, cultures were incubated for a final 5 min with 10 µM Hoechst 33342, then spun down and resuspended in a small volume of phosphate-buffered saline. Cells were then immediately visualized with a 100× plan-apochromat objective using a GFP filter set or a DAPI filter set for Hoechst 33342. 

### 2.7. Protein Analysis

Proteins for Western blotting were extracted using NaOH/β-mercaptoethanol (βME). Culture aliquots were spun down and resuspended in 1 mL of 250 mM NaOH and 1% βME and incubated for 10 min on ice. Then, trichloroacetic acid was added to 5% and the cells were incubated for another 10 min, at least, on ice. The precipitate was pelleted, in a refrigerated Eppendorf centrifuge; the pellet was washed with cold 100% acetone, dried, and then resuspended in gel loading buffer with 4% βME, 40 µL/OD_600_ unit. The samples were run on a 6% polyacrylamide gel, transferred to a PVDF membrane, and reacted in TBST (Tris-buffered saline + 0.1% Tween 20) with the anti-Myc 9E10 monoclonal antibody (Invitrogen, Carlsbad, CA, USA, 1:500) and a secondary horseradish peroxidase-conjugated anti-mouse antibody (Sigma-Aldrich, 1:10,000). The membranes were reacted with an Amersham ECL Plus kit (GE Healthcare, Chicago, IL, USA) and visualized with a FUSION FX7 Edge imaging system (Witec AG, Sursee, Switzerland).

## 3. Results

### 3.1. BPS Induces Filamentous Growth

When analyzing the pathway of heme-iron acquisition in *C. albicans*, we make extensive use of an experimental system where 1–2 mM BPS is used to chelate iron in the medium, and growth is recovered by addition of hemin or hemoglobin to the medium [29,30]. We noticed that under these conditions, cells that recovered growth in the presence of hemoglobin (or hemin) were largely hyphal after two days (Figure 1A). There have been some reports of hemin causing hyphal morphogenesis, particularly at high concentrations [31,32], but in these reports, BPS was already suggested to be an important contributing factor [32]. Hemoglobin has also been shown to cause hyphal morphogenesis [33], however, at much higher concentrations (1 mg/mL = 15 µM) than our own typical working concentrations (0.25–1 µM). To clarify the role of hemoglobin vs. BPS while maintaining conditions of utilization of heme as an iron source, we used an alternative iron chelator, ferrozine [34], which does not completely inhibit growth of wild-type strains but does completely inhibit growth of a *ccc2^−/−^* mutant, defective in high-affinity iron uptake [35]. Comparison of the morphologies of the *ccc2^−/−^* strain grown in the presence of hemoglobin and either BPS or ferrozine as iron chelators indicated that only BPS efficiently induced filament formation (Figure 1A), indicating that utilization of heme as an iron source is not the trigger for hyphal morphogenesis. Furthermore, under these conditions, hemoglobin alone, even at much higher concentrations, did not induce hyphal morphogenesis (Figure 1A). Lastly, to refute the remaining possibility that only the combination of BPS with hemoglobin can induce hyphal growth, we tested the effect of rescuing BPS inhibition with the siderophore ferrichrome pre-loaded with iron [36]. While growth rescue was only partial, the cells exhibited a mixture of elongated hyphae and short, germ tube-like hyphae, supporting the notion that BPS is itself an inducer of hyphal morphogenesis (Appendix A).

To further characterize the effect of BPS on filamentation, we tested different BPS concentrations and quantitated growth and filamentation in the absence and presence of hemoglobin at two different concentrations. As shown in Figure 1B, at low BPS concentrations, no filamentation was detected, but at concentrations that were inhibitory in the absence of heme-iron, namely above 1 mM, extensive filamentation could be measured after 2 days of incubation. In the iron uptake mutant *ccc2^−/−^*, while, as expected, a higher sensitivity to BPS was observed in the absence of hemoglobin, in its presence, the same extent of filamentation was detected at the same BPS concentrations (Figure 1B). Furthermore, no significant difference in filamentation was measured at 0.25 µM vs. 2.5 µM hemoglobin. Together, these results further confirm that, under these conditions, BPS is directly inducing filamentation.

### 3.2. Identification of Factors Required for BPS-Induced Hyphal Morphogenesis

#### 3.2.1. Screening of Transcription Factor Mutants

Different transcription factors and signal transduction pathways have been identified that are involved in hyphal morphogenesis in *C. albicans*, depending on the induction signals [7,8,37]. In order to identify mutants required for BPS-induced filamentation, we scanned the Homann library of transcription factor mutants [12], as well as a selection of mutants from our lab stock, including hgc1^−/−^, efg1^−/−^, cph1^−/−^, and ume6^−/−^, for defects in hyphal morphogenesis [19]. In a first pass, the clones were grown in 96-well plates in YPD + 2 mM BPS, 0.25 µM hemoglobin, for 48 h at 30°. Selected clones were re-checked in individual tubes (Figure 2). Mutants that showed a strong defect in hyphal formation include the transcription factors efg1^−/−^, ume6^−/−^, rob1^−/−^ and the hyphal-specific cyclin hgc1^−/−^, whereas sfl2^−/−^, tec1^−/−^, ndt80^−/−^, and cph1^−/−^ exhibited a partial defect. cph2^−/−^ and hap43^−/−^ are examples of transcription factor mutants that were unaffected in BPS-induced hyphal morphogenesis. 

#### 3.2.2. Unbiased Selection for Non-Filamenting Mutants

In order to identify additional factors in the BPS-induced filamentation pathway, we performed an unbiased, forward genetic screen using a transposon-mutagenized pool of haploid cells [16]. The pool was first enriched for non-filamenting mutants by repeated removal of slower-sedimenting cells over several subcultures, as described in Methods, and the enriched pool was plated on BPS and hemoglobin plates and visually scanned for clones exhibiting reduced hyphal formation [19]. Less-filamenting clones were then genotyped for the locus of transposon insertion. A total of 48 clones were characterized at the genotype level (Appendix A). A total of 15 insertions were in unique loci, mostly in intergenic regions, whereas the other 35 were represented between 2 and 12 hits within the coding region or the promoter region of 7 genes: *FLO8* (12), *EFG1* (5), *orf19.384* (5), *YAK1* (4), *UME6* (3), *MSH6* (2), and *EHT1* (2). Efg1 and Ume6 are transcription factors that were also identified in our directed screen (Figure 2). Flo8 is another known regulator of hyphal growth [38], which could not have been identified in our previous screen because it is not represented in the Homann library. Msh6 is a DNA mismatch repair gene homolog, and Eht1 is a putative fatty acid biosynthesis enzyme, neither of which had previously been linked to hyphal morphogenesis. 

### 3.3. Yak1 and Orf19.384

Of the last two genes identified by multiple hits, *YAK1* and *orf19.384*, the former encodes a kinase belonging to the DYRK family of dual-specificity kinases, conserved from fungi to mammals [39,40], which had been previously implicated in the initiation and maintenance of hyphal growth [41]. Orf19.384, in contrast, has not been characterized before. The analysis of the Orf19.384 sequence in the Interpro database [42] indicated that it is an ortholog of the conserved WDR68/DCAF7 (WD-40 Repeat 68/DDB1- and CUL4-Associated Factor 7) proteins, associated with developmental pathways in plants and animals [43,44,45]. Strikingly, WDR68/DCAF7 was found to interact with the Yak1 homologs DYRK1A and DYRK1B in animal cells [46,47,48]. 

#### 3.3.1. Confirmation of the Phenotypes in the Standard Diploid Background

To confirm the role of these two genes in BPS-induced filamentation in *C. albicans*, we deleted them in the standard diploid strain and tested the phenotype of the heterozygous and homozygous mutants grown in BPS and hemoglobin. As shown in Figure 3A, the *YAK1^+/−^* heterozygote already exhibited a partial defect in filamentation, whereas the *yak1^−/−^* homozygote was completely defective in filamentation. For *ORF19.384*, in contrast, the heterozygote was as filamentous as the wild-type, but the *orf19.384^−/−^* homozygote was profoundly defective in filamentation. We confirmed that the reduced filamentation in the mutants is not due to reduced growth, e.g., to an inability to utilize hemoglobin (Appendix A).

Since Goyard et al. have shown that the *yak1^−/−^* mutant is defective in filamentation in Lee’s medium [41], we also tested the heterozygous and homozygous *YAK1* and *ORF19.384* mutant strains in this medium and monitored hyphal formation after 5 h and 24 h (Figure 3B). We confirmed that the *yak1^−/−^* mutant is completely defective in hyphal formation in Lee’s medium, and we found that the heterozygote is already partially defective, similar to the phenotype in BPS-induced filamentation. For *ORF19.384*, in the heterozygote, hyphae already appeared shorter and less prominent, whereas in the homozygous *orf19.384^−/−^* mutant, very few hyphae were visible.

#### 3.3.2. Subcellular Localization

To further analyze the interaction between Yak1 and Orf19.384, we fused the *YAK1* and *ORF19.384* open reading frames to GFP and expressed the fusion proteins under their native promoter in wild-type cells or in cells lacking the other partner. As shown in Figure 4, wild-type cells show cytoplasmic and nuclear localization of both Yak1-GFP and Orf19.384-GFP, with a higher concentration in the nucleus of both proteins. However, Yak1-GFP expression in the *orf19.384^−/−^* mutant showed lower levels overall and remained visible only in the nucleus. Conversely, Orf19.384-GFP expressed in *yak1^−/−^* cells was still visible in the cytoplasm but lost its specific nuclear localization. The growth of the cells in synthetic medium yielded a very similar picture (Appendix A). Thus, Yak1 and Orf19.384 affect each other’s levels or localization.

#### 3.3.3. Sfl1 Is a Candidate Substrate of Yak1/Orf19.384

Based on the proposed DYRK1/Yak1 substrate consensus RPX(S/T)P [49], we scanned the *C. albicans* proteome for potential Yak1 targets. Among some 60 proteins having one potential target site each (Appendix A) is Sfl1, a transcription factor that was identified as a suppressor of hyphal morphogenesis. In particular, Sfl1 was proposed to be antagonistic to Flo8 [50] and to Sfl2 [51], and to co-bind its targets with Efg1 and/or Ndt80 [51], all factors that were identified in our screens. We therefore tested whether Sfl1 could be a target of Yak1 together with Orf19.384. To this end, we tagged Sfl1 with a 13xMyc tag at its C-terminus in wild-type, *yak1^−/−^* and *orf19.384^−/−^* cells. As shown in Figure 5, in two different media, Sfl1 migration was slower in the wild-type cells than in the *yak1^−/−^* or *orf19.384^−/−^* cells, while the two mutant strains exhibited a similar band pattern. Importantly, the proteins were analyzed in yeast growth conditions (SC medium) as well as in hyphal-inducing conditions (Lee’s medium). The observation that even in SC medium, where the morphologies of the wild-type and mutant strains are indistinguishable, the modification pattern of Sfl1 was very different in the wild-type vs. the mutants indicates that it does not represent an indirect effect of cell morphology on Sfl1 modification. This observation therefore supports the possibility that Yak1 and Orf19.384 function together to modify this substrate. 

## 4. Discussion

### 4.1. BPS and Filamentation

We have shown here that the iron chelator BPS induces, at high concentrations, hyphal morphogenesis in *C. albicans*. Since high BPS concentrations preclude growth by withholding essential iron from the cells, this filamentation is detectable only in the presence of hemoglobin (or hemin—our unpublished results), which restores growth of *C. albicans* cells by serving as an alternative iron source, thereby enabling the cells to manifest the BPS-induced hyphal morphology [35,52]. 

Notably, the addition of BPS had been previously shown by Hameed et al. to induce hyphal morphogenesis [53]. These authors interpreted the results as showing that iron deprivation, rather than BPS per se, was the proximal inducer of filamentation, based on the filamentous phenotype of the *ftr1* and *ccc2* high-affinity iron uptake mutants, even in the absence of BPS. A more recent report also found that BPS can induce hyphae formation on plates, dependent on a new iron regulator, the transcription factor Irf1 [54]. One factor that distinguishes these experimental systems from ours is that the BPS concentration used, 150 µM, was much lower and does not, in our hands, induce filamentation (Figure 1B). This discrepancy is probably due, in one case, to temperature. While we grew the cells at 30 °C, Hameed et al. used 37 °C, a condition known by itself to induce, or strongly contribute to, hyphal morphogenesis [6]. In the second case, filamentation was only detected after prolonged incubation on plates [54]. If iron withholding by BPS were inducing filamentation in our system, then the higher hemoglobin concentrations would have reduced filamentation. Thus, it is likely that Hameed et al. describe hyphal morphogenesis induced by the combination of high temperature and iron limitation, whereas we found a BPS-specific hyphal induction mechanism. Further indication that the two protocols induce filamentation by different pathways comes from the observation that while the Homann collection *hap43^−/−^* mutant is defective in hyphal induction in 150 µM BPS [12], this mutant is fully filamentous in 2 mM BPS at 30 °C (Figure 2). 

Since hemoglobin had been shown to induce hyphal morphogenesis on its own, albeit at a higher concentration of 15 µM [33], it was important to show that in our experiments, the hemoglobin was not responsible for this phenotype. We did this in several ways: by showing that an alternate iron chelator, ferrozine, does not induce filamentation in the presence of the same hemoglobin concentrations, even in cells that are highly sensitive to iron limitation; and by showing that, by varying the BPS and hemoglobin concentrations, the extent of filamentation was strongly dependent upon BPS concentration but not hemoglobin concentration. Thus, neither the presence of hemoglobin, nor iron starvation per se or the utilization of hemoglobin as iron source, can explain the BPS-induced filamentation phenotype. What, then, is inducing this phenotype? We did notice that at high BPS concentrations, i.e., above 1 mM, growth becomes progressively inhibited until it is completely inhibited at 4 mM, even in the presence of hemoglobin (Figure 1). This suggests that at a high enough concentration, BPS exerts effects on the cell that go beyond the chelation of iron in the environment. It is possible that, at a high enough concentration, BPS can penetrate the cell and interfere with cellular pathways via chelation of intracellular iron or by other means, thereby inducing the switch to hyphal morphogenesis. 

### 4.2. Mutants Defective in BPS-Dependent Filamentation

To try to understand the mechanism of action of BPS on cellular morphogenesis, we carried out a screen of mutants of known filamentation factors as well as an unbiased, forward genetics screen, looking for mutants exhibiting reduced filamentation in the presence of BPS. Taken together, the two screens identified a set of six transcription factors, namely Efg1, Ndt80, Rob1, Flo8, Sfl2, and Cph1, that are required for BPS-induced filamentation. Efg1 and Ndt80 are central transcription factors of both hyphal morphogenesis [55,56] and biofilm formation [57], and are usually found bound together to the same promoters [58]. Cph1 [23], Flo8 [38], and Sfl2 [59,60] were identified as positive regulators of hyphal morphogenesis as well: Flo8 interacts directly with Efg1 [38] and Sfl2 with Efg1 and Ndt80, and their binding sites usually co-occur on target promoters [51]. Rob1 was associated mainly with biofilm formation [57], and strikingly, it, together with Efg1, Ndt80, Flo8, and Brg1, were proposed to be among the core genes in the transcription network that regulates biofilm formation [57,61]. Thus, our screens have identified a coherent subset of interacting transcription factors involved in hyphal morphogenesis and in the related pathway of biofilm formation.

### 4.3. Yak1 and Orf19.384

Two additional genes that came up repeatedly in the transposon-mutagenized pool were *YAK1* and *ORF19.384. YAK1* was previously shown to affect hyphal morphogenesis and hyphal-specific transcription in *C. albicans* [41]. In *S. cerevisiae,* Yak1 is regulated by the protein kinase A (PKA) pathway, and it affects i.a. adhesive growth by activating expression of the flocculin gene *FLO11*, itself a target of *S. cerevisiae* Flo8 [62]. *C. albicans* Yak1 was recently shown to also function within the PKA pathway and to require the transcription factors Efg1 and Flo8 for the induction of filamentation [63]. In addition, *C. albicans* Yak1 can be inhibited by a lactobacillus metabolite, 1-ABC, that inhibits hyphal morphogenesis, and analysis of genomic suppressors of 1-ABC-mediated suppression of filamentation pointed to Rob1 as a possible Yak1 target [64].

Yak1 belongs to the conserved dual-specificity tyrosine-phosphorylated and regulated kinase (DYRK) family [39,40]. DYRK kinases are involved in the regulation of cellular growth and differentiation in invertebrates and vertebrates [48,65,66]. In several animal models, DYRK1-type kinases were found to interact with another conserved family of proteins, the WD40 repeat protein family WDR68/DCAF7 [46,47,48]. One proposed role for these proteins is to mediate the interaction of the DYRK kinases with their substrates [67]. The fact that Orf19.384 is the single apparent ortholog of WDR68 in *C. albicans* suggests that it may function together with Yak1. Furthermore, the single apparent Orf19.384/WDR68 ortholog in *S. cerevisiae*, Ypl247c, was shown in two different proteomics screens to physically interact with the *S. cerevisiae* Yak1 ortholog [68,69].

On the assumption that Yak1 and Orf19.384/Wdr68 interact as well, we first tested whether they affect each other’s subcellular localization. We found that both proteins are normally localized in the cytosol and concentrated in the nucleus, but in the absence of Yak1, Orf19.384 lost its nuclear localization. Conversely, in the absence of Orf19.384, Yak1 was only detectable in the nucleus. This, however, could be due to overall lower amounts of the Yak1 protein in cells lacking Orf19.384, rather than to relocalization from the cytosol to the nucleus (Figure 4). In any case, we find that Yak1 and Orf19.384 affect each other’s localization and/or levels. Interestingly, similar to the dependence of Orf19.384/Wdr68 on Yak1 for its nuclear accumulation, human WDR68 was found to depend on the Yak1 homolog DYRK1A for its nuclear accumulation as well [46]. 

To further support the conjecture that Yak1 and Orf19.384/Wdr68 function together in *C. albicans*, we attempted to identify a potential Yak1 target. Among some 60 proteins that contain the DYRK consensus RPX(S/T)P, Sfl1 stood out, based on its known relationship with many of the other genes identified in our screen. Sfl1 is a repressor of hyphal morphogenesis [50,70], which was suggested to antagonize Flo8 [50] as well as Sfl2 [51]. It was suggested to be centrally involved in the formation of biofilm-like microcolonies on an oral mucosa model, together with Sfl2, Rob1, and Ndt80 [71], and to suppress hyphal formation in an acidic medium [72]. In our hands, Sfl1 was indeed found to be differentially modified in wild-type vs. *yak1^−/−^* or *orf19.384^−/−^* cells, even in normal yeast growth medium. This supports the notion that Yak1 and Orf19.384 function together in *C. albicans* and links them to the transcription network defined by the other non-filamenting mutants identified here.

Taken together, our data suggest a deep conservation of the function of the DYRK-WDR68/DCAF7 and Yak1-Orf19.384 complexes across the animal and fungal kingdoms. Based on the sequence and functional homology of *ORF19.384* to its mammalian homologs, we suggest renaming it *WDR68*.

## 5. Conclusions

We have identified a new protocol for hyphal induction in *C. albicans*: exposure to high BPS concentrations in the presence of hemoglobin as an alternative iron source. A genetic analysis of this hyphal induction pathway revealed a subset of hyphal- and biofilm-specific transcription factors that were known to interact physically and/or functionally, as well as a conserved protein kinase complex, Yak1-Orf19.386/Wdr68, whose homologs are involved in cellular differentiation in animals. Considering that the dimorphic switch from yeast to hyphal morphology represents a form of cellular differentiation, we conclude that the conservation of structure as well as function of this complex extends to the fungal kingdom. We propose that fungi represent a useful model system for the study of the function, mechanism, and regulation of DYRK kinases.

## Figures and Tables

**Figure 1 jof-10-00083-f001:**
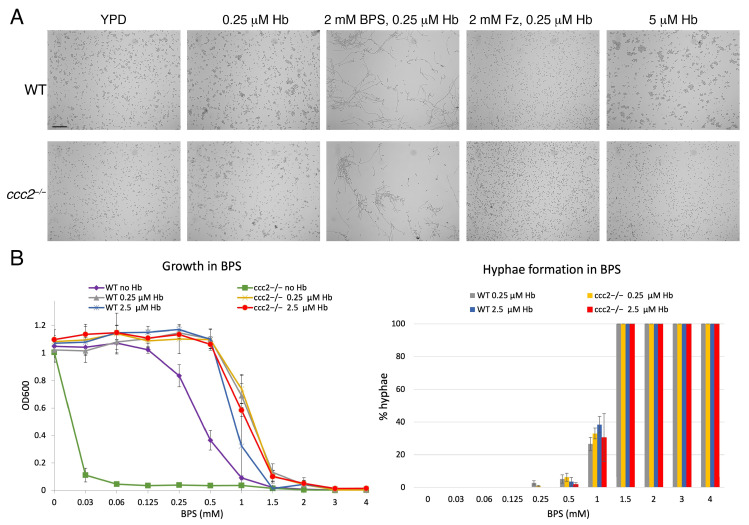
Induction of filamentous growth in the presence of BPS. (**A**). Wild-type (KC2) or *ccc2^−/−^* mutant cells (KC68) were grown for 48 h at 30° in YPD medium in aerated test tubes with the indicated additions: 2 mM BPS, 0.25 µM hemoglobin (Hb), or 2 mM ferrozine (Fz). Scale bar = 100 µm. (**B**). BPS sensitivity of growth and hyphae formation in the wild-type (KC2) or *ccc2^−/−^* mutant cells (KC68). The strains were diluted to OD_600_ = 0.0001 in the indicated media and grown for 48 h in 96-well plates at 30° while shaking. The optical densities represent the average of three different cultures. The error bars indicate the standard deviations. For morphology, a sample was placed under the microscope at 10X magnification, and fields were scanned until at least 100 cells were counted for each condition.

**Figure 2 jof-10-00083-f002:**
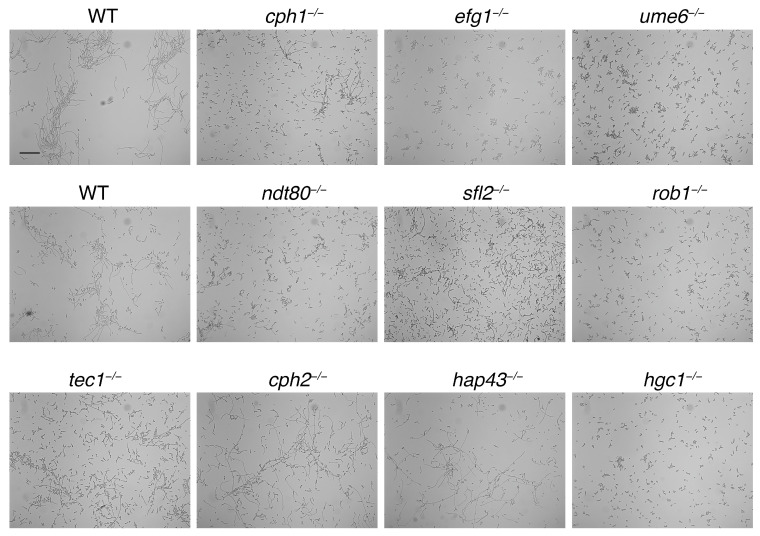
Screen for mutants that are defective in BPS-induced filamentation. Selected strains were incubated in YPD with 2 mM BPS and 0.25 µM hemoglobin at 30 °C for 48 h. The strains shown include mutants defective in BPS-induced filamentation as well as mutants that do not show a phenotype under these conditions. Top row: selected strains from our lab stock (KC2, KC148, KC149, and KC445). Middle and bottom row: selected strains from the Homann library [12], except the *hgc1^−/−^* mutant [26]. The scale bar is 100 µm. Quantitation of the percentage hyphae is shown in Appendix A.

**Figure 3 jof-10-00083-f003:**
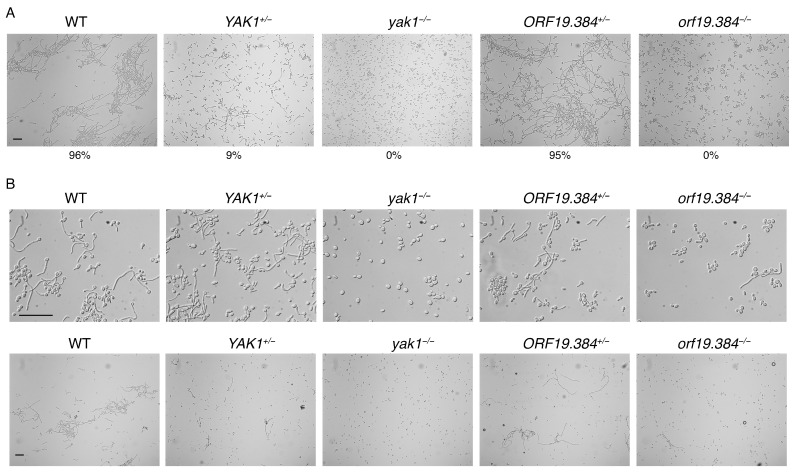
Phenotype of the *yak1* and *orf19.384* mutants. The indicated strains, wild-type (KC1175), *YAK1^+/−^* (KC1336), *yak1^−/−^* (KC1338), *ORF19.384^+/−^* (KC1340), and *orf19.384^−/−^* (KC1363), were incubated (**A**) in YPD with 2 mM BPS and 0.25 µM hemoglobin at 30 °C for 48 h or (**B**) in Lee’s medium at 37 °C for 5 h (top) or 24 h (bottom). The numbers indicate the percentage of hyphal cells in the BPS-induced cultures. The scale bars are 50 µm.

**Figure 4 jof-10-00083-f004:**
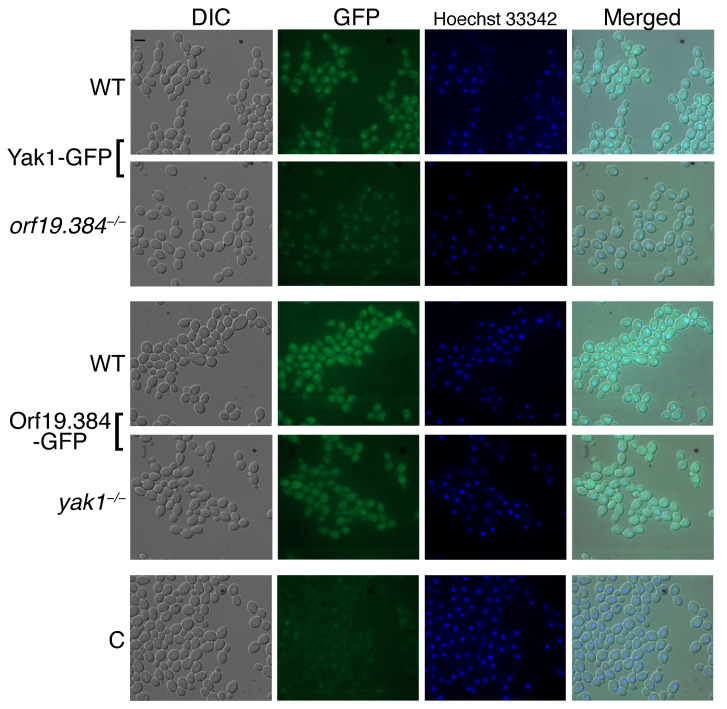
Effect of deletion of *YAK1* or *ORF19.384* on expression and localization of their partner protein. Wild-type (KC1669) and *orf19.384^−/−^* (KC1672) cells expressing *YAK1* fused to GFP and wild-type (KC1670) and *yak1^−/−^* (KC1671) cells expressing *ORF19.384* fused to GFP were grown to log phase in YPD medium at 30 °C, incubated for 5 min with Hoechst 33342 for nuclear staining, and visualized by DIC and epifluorescence microscopy. C = untagged control strain KC1175. Scale bar = 5 µM.

**Figure 5 jof-10-00083-f005:**
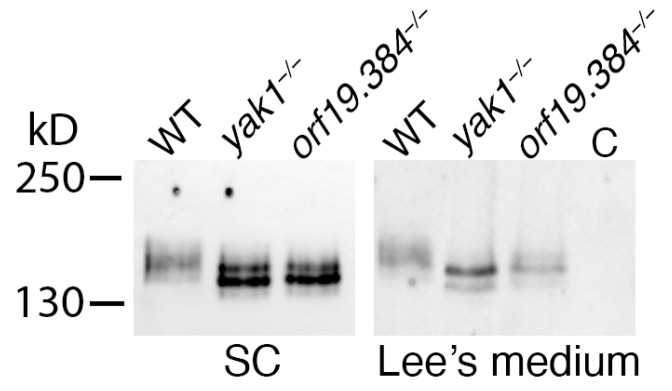
Reduced modification of Sfl1-13xMyc in *yak1^−/−^* and *orf19.384^−/−^* cells. Wild-type (KC1543), *yak1^−/−^* (KC1544), and *orf19.384^−/−^* (KC1545) cells expressing an *SFL1* allele fused to the 13xMyc epitope tag were grown 4 h in SC medium or 5 h in Lee’s medium at 30 °C, and the protein extracts were submitted to Western blotting and reacted with the anti-Myc 9E10 antibody. C indicates the extract of a non-tagged control strain (KC274).

**Table 1 jof-10-00083-t001:** List of *Candida albicans* strains.

Name	Genotype/Strain Number	Origin
Diploid strains		
KC2 = CAI4	*ura3*Δ*::imm434/ura3*Δ*::imm434*	[21]
KC148 = JKC18	*ura3*Δ*::imm434/ura3*Δ*::imm434 cph1*Δ*/cph1*Δ	[22]
KC149 = HLC52	*ura3*Δ*::imm434/ura3*Δ*::imm434 efg1*Δ*/efg1*Δ	[23]
KC274 = SN148	*ura3*Δ*::imm434/ura3*Δ*::imm434, his1*Δ*/his1*Δ*, leu2*Δ*/leu2*Δ*, arg4*Δ*/arg4*Δ	[24]
KC445	*ura3*Δ*::imm434/ura3*Δ*::imm434 ume6*Δ*::hisG/ume6*Δ*::hisG*	[25]
KC532	KC274 *hgc1*Δ*::HIS1/hgc1*Δ*::ARG4*	[26]
KC1175	KC274 *ADE2/ade2::URA3*	This work
KC1336	KC274 *YAK1/yak1*Δ*::hisG-URA3-hisG*	[19]
KC1337	KC274 *YAK1/yak1*Δ*::hisG*	[19]
KC1338	KC274 *yak1*Δ*::hisG/yak1*Δ*::hisG-URA3-hisG*	[19]
KC1339	KC274 *yak1*Δ*::hisG/yak1*Δ*::hisG*	[19]
KC1340	KC274 *ORF19.384/orf19.384*Δ*::hisG-URA3-hisG*	[19]
KC1341	KC274 *ORF19.384/orf19.384*Δ*::hisG*	[19]
KC1363	KC274 *orf19.384*Δ*::hisG/orf19.384*Δ*::hisG*	[19]
KC1381	KC274 *orf19.384*Δ*::hisG/orf19.384*Δ*::hisG*	[19]
KC1543	KC274 *SFL1-13xMyc URA3*	This work
KC1544	KC1339 *SFL1-13xMyc URA3*	This work
KC1545	KC1381 *SFL1-13xMyc URA3*	This work
KC1669	KC274 *YAK1/YAK1-GFP URA3*	This work
KC1670	KC274 *ORF19.384/ORF19.384-GFP URA3*	This work
KC1671	KC1339 *ORF19.384/ORF19.384-GFP URA3*	This work
KC1672	KC1381 *YAK1/YAK1-GFP URA3*	This work
Haploid strains		
KC1139	*ura3*Δ *ade2::Ds-NAT1 NEU5tl::AcTPase4x URA3*	[16]
KC1140	*ura3*Δ	[16]

## Data Availability

All the data appear in the manuscript figures and attached Appendix A.

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
