# Peer review of "Genetic Analysis of Candida albicans Filamentation by the Iron Chelator BPS Reveals a Role for a Conserved Kinase—WD40 Protein Pair"

_jof, 2024, doi:10.3390/jof10010083_

Round 1
Reviewer 1 Report
Comments and Suggestions for Authors
This is a potentially interesting study that investigates the filamentation of C. albicans induced by combined treatment with the ferrous iron selective chelator BPS and heme. By screening available mutant libraries, the authors were able to demonstrate a role for the Yak1 kinase and its putative partner, orf9.384 in this very specific form of filamentation. The findings are intriguing and the experiments rigorous. However, the study as a whole is confusing, because the authors are combining two distinct agents, BPS and heme, and it is unclear as to whether a disruption in Fe homeostasis is involved or whether combining these agents causes some other form of stress unrelated to Fe. Fe starvation has already been documented to induce filamentation in other published studies. In the current study, the authors argue that BPS+heme is not inducing filamentation through Fe starvation and is therefore novel, but there were no experiments that investigate whether the cells are Fe starved or not.
Suggestions for improvement:
1) The authors conclude that the presence of heme is overcoming Fe starvation induced by BPS, but there is no supporting data. There are numerous reliable assays of Fe starvation that the authors can use, including various mRNA markers of Fe starvation stress, whole cell ferric reductase activity and biochemical assays of total cellular Fe.
2) Another interpretation of the findings is that filamentation is induced when heme becomes the sole source of cell Fe. With heme alone, cells can also take up Fe through the reductive Fe uptake pathway, but with BPS+heme, cells are forced to only use heme. The authors do have a ferrozine + heme control, but unfortunately they only tested one dose of ferrozine and since ferrozine and BPS are not chemically equivalent, they likely have different dose optimums. To demonstrate that heme as a sole Fe source is not the cause of filamentation, the authors can substitute heme with a siderophore, or siderophore like molecule. If BPS+siderophore also induces filamentation, then the authors can conclude that heme is irrelevant and BPS indeed on its own is responsible.
Minor comments:
3) In several places the authors state that BPS is inducing filamentation, which can confuse the readers because the data only shows BPS+heme inducing filamentation, not BPS alone. Examples include the title and the last sentence of the first paragraph of the introduction. Unless the authors provide additional mechanistic data (e.g., suggestions 1 and 2 above), they need to modify the text throughout, including the title, to clearly state that filamentation is induced by the combination of BPS and heme.
4) Fig. 1A needs the no treatment control image.
5) In the discussion the authors state: One factor that distinguishes these experimental systems from ours is that the BPS concentration used, 150 µM, was much lower, and does not, in our hands, induce filamentation (Fig. 1). But there is no filamentation experiment in Fig. 1 that shows a dose response of BPS alone and hyphae. The only dose response for BPS alone is in 1B which is a test of growth, not filamentation.
6) Can the authors give orf9.384 a three letter one number name? When this study is published, it could be helpful for gene annotation.
Author Response
We thank the reviewer for their useful comments. Please see our point by point response to the comments in the attachment. We have modified the manuscript accordingly. We hope you will find the revised manuscript acceptable for publication.

Reviewer 2 Report
Comments and Suggestions for Authors
In the manuscript by Pinsky and Kornitzer, the Authors report an involvement of a DYRK family kinase Yak1 and a WD40 repeat protein Orf19.384 in filamentous growth of Candida albicans. The Authors first establish that the iron chelating chemical BPS induces filamentation in the presence of hemoglobin, which at relatively high concentrations of BPS likely happens through a mechanism that is (at least partially) independent of iron chelation. The Authors then leverage this phenomenon and set up genetic screens to define novel genes involved in filamentation in C. albicans. Among several genes that have been previously described as involved in filamentous growth, the Authors uncover Orf19.384 and the Yak1 kinase, a pair for which homologous proteins have been linked to morphogenesis in animals. As the Authors note, Yak1 has been previously linked to hyphal growth in S. cerevisiae and C. albicans and in S. cerevisiae a homologue of Orf19.384 has been shown to physically interact (directly or indirectly) with the homologue of Yak1. The novel aspect of this study is that in C. albicans the two proteins are also involved in filamentation and the fact that they influence each other’s subcellular localization. Another novel aspect shown is that the two proteins influence modification (likely phosphorylation) of Sfl1, a suppressor of hyphal morphogenesis. While this study leaves many open questions, it constitutes a well-executed initial report on the Yak1/Orf19.384 pair’s involvement in hyphal growth of C. albicans. The following suggestions should help to further improve this manuscript.
1. Ln 71: space is missing between YAK1 and open.
2. Ln 98: suggest replacing "that" with "than"
3. How many independent WT strains were tested for the effects of BPS+hemoglobin? Was only one specific genetic background tested?
4. Ln 175: C. albicans should be italic
5. Ln 176: is this to find mutants that are specific to BPS? - not necessarily – they may not be specific to BPS.
6. Ln 177-185: I suggest the Authors apply some form of quantification for this assessment as it is not evident based on images that some mutants show a partial versus more substantial defect.
7. Ln 228: The second "due" seems to be in a wrong place/not needed here
8. Ln 301: was the strain genetic background the same in all three studies?
9. Ln 369: The Authors conclude: "in the absence of Orf19.384, Yak1 was only detectable in the nucleus". Data shown in Figure 4 do not support this conclusion - there is still residual Yak1-GFP fluorescence in the cytoplasm in the mutant. Quantification of the data could help here.
10. Testing if there a change in localization of either of the proteins upon addition of BPS+hemoglobin could be informative?
Author Response
We thank the reviewer for their useful comments. Please see our point by point response to the comments in the attachment. We have modified the manuscript accordinglyd. We hope you will find the revised manuscript acceptable for publication.

Round 2
Reviewer 1 Report
Comments and Suggestions for Authors
The authors have adequately addressed the concerns of the previous review and no additional revisions are required